# Fast Adaptation in Generative Models with Generative Matching Networks

**Sergey Bartunov**[†] **& Dmitry P. Vetrov**[‡]
National Research University Higher School of Economics (HSE)[†‡]
Moscow, Russia

Yandex[‡]
Moscow, Russia

## Abstract

Despite recent advances, the remaining bottlenecks in deep generative models are necessity of extensive training and difficulties with generalization from small number of training examples. Both problems may be addressed by conditional generative models that are trained to adapt the generative distribution to additional input data. So far this idea was explored only under certain limitations such as restricting the input data to be a single object or multiple objects representing the same concept. In this work we develop a new class of deep generative model called generative matching networks which is inspired by the recently proposed matching networks for one-shot learning in discriminative tasks and the ideas from meta-learning. By conditioning on the additional input dataset, generative matching networks may instantly learn new concepts that were not available during the training but conform to a similar generative process, without explicit limitations on the number of additional input objects or the number of concepts they represent. Our experiments on the Omniglot dataset demonstrate that generative matching networks can significantly improve predictive performance on the fly as more additional data is available to the model and also adapt the latent space which is beneficial in the context of feature extraction.

## 1 Introduction

Deep generative models are currently one of the most promising directions in generative modelling. In this class of models the generative process is defined by a composition of conditional distributions modelled using deep neural networks which form a hierarchy of latent and observed variables. This approach allows to build models with complex, non-linear dependencies between variables and efficiently learn the variability across training examples.

Such models are trained by stochastic gradient methods which can handle large datasets and a wide variety of model architectures but also present certain limitations. The training process usually consists of small, incremental updates of networks' parameters and requires many passes over training data. Notably, once a model is trained it cannot be adapted to newly available data without complete re-training to avoid catastrophic interference (McCloskey & Cohen, 1989; Ratcliff, 1990). There is also a risk of overfitting for concepts that are not represented by enough training examples which is caused by high capacity of the models. Hence, most of deep generative models are not well-suited for rapid learning in one-shot scenario which is often encountered in real-world applications where data acquisition is expensive or fast adaptation to new data is required.

A potential solution to these problems is explicit learning of adaptation mechanisms complementing the shared generative process. In probabilistic modelling framework, adaptation may be expressed as conditioning the model on additional input examples serving as induction bias. Notable steps in this direction have been made by Rezende et al. (2016) whose model was able to condition on a single object to produce new examples of the concept it represents. Later, Edwards & Storkey (2016) proposed a model that maintained a global latent variable capturing statistics about multiple input objects which was used to condition the generative distribution. It allowed to implement the

fast learning ability, but due to the particular model architecture used the model was not well-suited to datasets consisting of several different concepts.

In this work we present Generative Matching Networks, a new family of conditional generative models capable of instant adaptation to new concepts that were not available at the training time but share the structure of underlying generative process with the training examples. By conditioning on additional inputs, Generative Matching Networks improve their predictive performance, the quality of generated samples and also adapt their latent space which may be useful for unsupervised feature extraction. Importantly, no explicit limitations on the conditioning data are imposed such as number of objects or number of different concepts which expands the applicability of one-shot generative modelling and distinguish our work from existing approaches. Our model is inspired by the attentional mechanism implemented in Matching Networks (Vinyals et al., 2016) previously proposed for discriminative tasks and the recent advances from meta-learning (Santoro et al., 2016). Our approach for adaptation is an extension of these ideas to generative modelling and it may be re-used in a variety of different models being not restricted to the particular architecture used in the paper. The source code for generative matching networks is available at `http://github.com/sbos/gmn`.

This paper is organized as follows. First, in section 2 we revisit the necessary background in variational approach to training generative models and mention the related work in conditional generative models. Then, in section 3 we describe the proposed generative model, it's recognition counterpart and the training protocol. Section 4 contains experimental evaluation of the proposed model as both generative model and unsupervised feature extractor in small-shot learning settings. We conclude with discussion of the results in section 5.

## 2 BACKGROUND

We consider the problem of learning a probabilistic generative model which can be expressed as a probability distribution $p(\mathbf{x}|\boldsymbol{\theta})$ over objects of interests $\mathbf{x}$ parametrized by $\boldsymbol{\theta}$. The major class of generative models introduce also *latent* variables $\mathbf{z}$ that are used to explain or generate an object $\mathbf{x}$ such that $p(\mathbf{x}|\boldsymbol{\theta}) = \int p(\mathbf{z}|\boldsymbol{\theta})p(\mathbf{x}|\mathbf{z}, \boldsymbol{\theta})d\mathbf{z}$ and assumed to be non-observable.

Currently, the common practice is to restrict the conditional distributions $p(\mathbf{z}|\boldsymbol{\theta})$ and $p(\mathbf{x}|\mathbf{z}, \boldsymbol{\theta})$ to tractable distribution families and use deep neural networks for regressing their parameters. The expressive power of deep non-linear generative models comes at a price since neither marginal distribution $p(\mathbf{x}|\boldsymbol{\theta})$ can be computed analytically nor it can be directly optimized in a statistically efficient way. Fortunately, intractable maximum likelihood training can be avoided in practice by resorting to adversarial training (Gutmann & Hyvärinen, 2012; Goodfellow et al., 2014) or variational inference framework (Kingma & Welling, 2013; Rezende et al., 2014) which we consider further.

### 2.1 TRAINING GENERATIVE MODELS WITH VARIATIONAL INFERENCE

Recent developments in variational inference alleviate problems with maximizing the intractable marginal likelihood $\log p(\mathbf{x}|\boldsymbol{\theta})$ by approximating it with a lower bound (Jordan et al., 1999):

$$\log p(\mathbf{x}|\boldsymbol{\theta}) \geq \mathcal{L}(\boldsymbol{\theta}, \boldsymbol{\phi}) = \mathbb{E}_q \left[ \log p(\mathbf{x}, \mathbf{z}|\boldsymbol{\theta}) - \log q(\mathbf{z}|\mathbf{x}, \boldsymbol{\phi}) \right] = \log p(\mathbf{x}|\boldsymbol{\theta}) - \mathrm{KL}(q||p(\cdot|\mathbf{x}, \boldsymbol{\theta})). \quad (1)$$

Tightness of the bound is controlled by the recognition model $q(\mathbf{z}|\mathbf{x}, \boldsymbol{\phi})$ which aims to minimize Kullback-Leibler divergence from the true posterior $p(\mathbf{z}|\mathbf{x}, \boldsymbol{\theta})$.

Similarly to the generative model, recognition model may also be implemented with the use of deep neural networks or other parameter regression which is known as *amortized inference* (Gershman & Goodman, 2014). Amortized inference allows to use a single recognition model for many training examples. Thus, it is convenient to perform training of the generative model $p(\mathbf{x}|\boldsymbol{\theta})$ by stochastic gradient optimization of variational lower bounds (1) corresponding to independent observations $\{\mathbf{x}_i\}_{i=1}^N$:

$$\sum_{i=1}^N \log p(\mathbf{x}_i|\theta) \geq \sum_{i=1}^N \mathbb{E}_q \left[ \log p(\mathbf{x}_i, \mathbf{z}_i|\boldsymbol{\theta}) - \log q(\mathbf{z}_i|\mathbf{x}_i, \boldsymbol{\phi}) \right] \to \max_{\boldsymbol{\theta}, \boldsymbol{\phi}}.$$

The clear advantage of this approach is its scalability. Every stochastic update to the parameters computed from only a small portion of training examples has an immediate effect for the whole

dataset. However, while a single parameter update may be relatively fast a large number of them is required to significantly improve generative or inferential performance of the model.

Hence, gradient training of generative models usually results into an extensive computational process which prevents from rapid incremental learning. In the next section we discuss potential solutions to this problem that allow to implement fast learning ability in generative models.

## 2.2 Adaptation in generative models

In probabilistic modelling framework the natural way of incorporating knowledge about newly available data is conditioning. One may design a model that being conditioned on the additional input data $\mathbf{X} = \mathbf{x}_1, \mathbf{x}_2, \ldots, \mathbf{x}_T$ represents a new generative distribution $p(\mathbf{x}|\mathbf{X}, \boldsymbol{\theta})$.

An implementation of this idea can be found in the model by Rezende et al. (2016). Besides many other attractive novelties such as using sophisticated attention and feedback components, the model was able to produce new examples of a concept that was missing at the training time but had similarities in the underlying generative process with the other training examples. The model supported an explicit conditioning on a single observation $\mathbf{x}'$ representing the new concept to construct a new generative distribution of the form $p(\mathbf{x}|\mathbf{x}', \boldsymbol{\theta})$.

The explicit conditioning when adaptation is performed *by the model* itself and and has to be learned is not the only way to propagate knowledge about new data. Another solution which is often encountered in Bayesian models is to maintain a *global* latent variable $\boldsymbol{\alpha}$ encoding information about the whole available dataset such that the individual observations are conditionally independent given it's value. The model then would have the following form:

$$p(\mathbf{X}|\boldsymbol{\theta}) = \int p(\boldsymbol{\alpha}|\boldsymbol{\theta}) \prod_{t=1}^{T} p(\mathbf{x}_t|\boldsymbol{\alpha}, \boldsymbol{\theta})d\boldsymbol{\alpha}. \tag{2}$$

The principal existence of such a global variable may be justified by the de Finetti's theorem (Diaconis & Freedman, 1980) under the exchangeability assumption. In the model (2), the conditional generative distribution $p(\mathbf{x}|\mathbf{X}, \boldsymbol{\theta})$ is then defined implicitly via posterior over the global variable:

$$p(\mathbf{x}|\mathbf{X}, \boldsymbol{\theta}) = \int p(\mathbf{x}|\boldsymbol{\alpha}, \boldsymbol{\theta})p(\boldsymbol{\alpha}|\mathbf{X}, \boldsymbol{\theta})d\boldsymbol{\alpha}. \tag{3}$$

Once there is an efficient inference procedure for the global variable $\boldsymbol{\alpha}$, fast adaptation of the generative model can be implemented straightforwardly.

There are several relevant examples of generative models with global latent variables used for model adaptation and one-shot learning. Salakhutdinov et al. (2013) combined deep Boltzmann machine (DBM) with nested Dirichlet process (nDP) in a Hierarchical-Deep (HD) model. While being a compelling demonstration of important ideas from Bayesian nonparametrics and deep learning, the HD model required an extensive Markov chain Monte Carlo inference procedure used both for training and adaptation. Thus, while Bayesian learning approach could prevent overfitting the fast learning ability still presents a challenge for sampling-based inference.

Later, Lake et al. (2015) proposed Bayesian program learning (BPL) approach for building a generative model of handwritten characters. The model was defined as a probabilistic program contained fine-grained specification of prior knowledge of the task such as generation of strokes and their composition into characters mimicking human drawing strategies. Authors used an extensive posterior inference as the training procedure and the conditioning mechanism (3) for generating new examples. The model was shown to efficiently learn from small number of training examples, but similarly to the HD model, sophisticated and computationally expensive inference procedure makes fast adaptation in BPL generally hard to achieve.

The recently proposed neural statistician model (Edwards & Storkey, 2016) is an example of deep generative model with a global latent variable (2). The model was trained by optimizing a variational lower bound following the approach described in section 2.1 but with an additional recognition model approximating posterior distribution over the global latent variable. Authors designed the recognition model to be computationally efficient and require only a single pass over data which consisted of extracting special features from the examples, applying to them a pooling operation (e.g. averaging) and passing the result to another network providing parameters of the variational approximation. This simple architecture allowed for the fast learning and guaranteed invariance to both data permutations and size of the conditioning dataset. However, experimentally the fast

learning ability in the model was evaluated only in the setting where all of the training examples represented the same single concept.

We argue that in order to capture more information about the conditioning data such as a number of different concepts a more sophisticated aggregation procedure must be employed. Moreover, a fixed parametric description is too restrictive for an accurate representation of datasets of varying size. This motivates us to combine the best of two worlds: nonparametric representation of data and fast inference with neural recognition models. We proceed with a description of the proposed model.

## 3 GENERATIVE MATCHING NETWORKS

Generative matching networks aim to model conditional generative distributions of the form $p(\mathbf{x}|\mathbf{X}, \boldsymbol{\theta})$. Similarly to other deep generative models we introduce a local latent variable $\mathbf{z}$. Thus the full joint distribution of our model can be expressed as:

$$p(\mathbf{x}, \mathbf{z}|\mathbf{X}, \boldsymbol{\theta}) = p(\mathbf{z}|\mathbf{X}, \boldsymbol{\theta})p(\mathbf{x}|\mathbf{z}, \mathbf{X}, \boldsymbol{\theta}). \tag{4}$$

We also maintain a recognition model approximating the posterior over the latent variable $\mathbf{z}$: $q(\mathbf{z}|\mathbf{x}, \mathbf{X}, \boldsymbol{\phi}) \approx p(\mathbf{z}|\mathbf{x}, \mathbf{X}, \boldsymbol{\theta})$.

In order to design a fast adaptation mechanism we have to make certain assumptions about relationships between training data and the new data used to condition the model. Thus we assume the homogeneity of generative processes for training and conditioning data up to some parametrization. One may think of this parametrization as specifying weights of a neural network defining a generative model. The generative process is assumed to have an approximately linear dependence on the parameters such that interpolation between parameters corresponding to different examples of the same concept can serve as good parameters for generating other examples. A similar assumption is used e.g. in the neural statistician model (Edwards & Storkey, 2016).

However, even if a single concept can be well embedded to a fixed parameter space, this does not imply that a diverse set of concepts will fit into the same parametrization. Hence we express the dependency on the conditioning data in a different way. Instead of embedding the whole conditioning dataset we use a special matching procedure that extracts relevant observations from $\mathbf{X}$ and interpolates between their descriptions allowing to generate and recognize similar observations.

### 3.1 BASIC MODEL

In the basic model, the prior over latent variables $p(\mathbf{z})$ is independent from conditioning data $\mathbf{X}$, e.g. a standard normal distribution. In order to generate a new object, a sample from the prior $\mathbf{z}$ and conditioning objects $\mathbf{X} = \mathbf{x}_1, \mathbf{x}_2, \ldots, \mathbf{x}_T$ are mapped into the *matching* space $\Phi$ where they are compared using a similarity function $\text{sim}(.,.)$ to form an *attention kernel* $a(\mathbf{z}, \mathbf{x})$. After that, the conditioning objects are interpolated in the *prototype* space $\Psi$ weighted according to the attention kernel. The resulting interpolation is then used to parametrize the generative process that corresponds to the sampled value of latent variable.

Formally, the described matching procedure can be described by the following equations:

$$\mathbf{r} = \sum_{t=1}^{T} a(\mathbf{z}, \mathbf{x}_t)\psi_L(\mathbf{x}_t), \quad a(\mathbf{z}, \mathbf{x}_t) = \frac{\exp(\text{sim}(f_L(\mathbf{z}), g_L(\mathbf{x}_t)))}{\sum_{t'=1}^{T} \exp(\text{sim}(f_L(\mathbf{z}), g_L(\mathbf{x}_{t'})))}. \tag{5}$$

After the vector $\mathbf{r}$ is computed, it is used as an input to a decoder, e.g. a deconvolutional network.

Functions $f_L$ and $g_L$ are used to map latent variables and conditioning objects, correspondingly, into the matching space $\Phi$. Since $\Phi$ is supposed to be a feature space that is good for discriminating between objects, $g_L$ can be implemented as a feature extractor suitable for the domain of observations, a convolutional network in our case. We found it sufficient to implement the function $f_L$ as a simple affine transformation followed by a non-linearity, because the latent variable itself is assumed to be an abstract object description. We also used a simple dot product as a similarity function between these vectors.

Function $\psi_L$ can also be considered as a feature extractor, although since the features useful to specify the generative process are not necessarily good for discrimination, it makes sense to represent

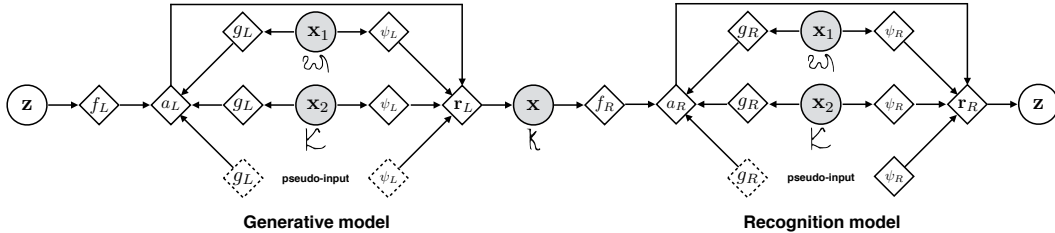

Figure 1: Structure of a basic generative matching network, see equation (5) in section 3.1 for the description of functions $f$, $g$ and $\psi$. Subscripts $L$ and $R$ denote conditional likelihood and recognition model correspondingly.

$\psi_L$ and $g_L$ differently. However, in our implementation $\psi_L$ was implemented as a convolutional network sharing most of the parameters with $g_L$ to keep the number of trainable parameters small.

We have described the basic matching procedure on the example of the conditional likelihood $p(\mathbf{x}|\mathbf{z}, \mathbf{X}, \boldsymbol{\theta})$. Although the procedure (5) is invoked in several parts of the model, each part may operate with it's own implementation of the functions, hence the subscript $\cdot_L$ used for the functions $f$, $g$ and $\psi$ is for likelihood part and below we use $\cdot_R$ to denote the recognition part.

The recognition model $q(\mathbf{z}|\mathbf{X}, \mathbf{x})$ uses the matching procedure (5) with the difference that the conditioning objects are being matched not with a value of latent variable, but rather with an observation $\mathbf{x}$. The feature extractor $f_R$ in this case can share most of the parameters with $g_R$ and in our implementation these functions were identical for matching in the recognition model, i.e. $g_R = f_R$ Moreover, since $g_L$ is also used to project observations into the space $\Phi$, we further re-use already defined functionality by setting $g_R = g_L$. We also shared prototype functions $\psi$ for all parts of our model although this is not technically required.

After the matching, interpolated prototype vector $\mathbf{r}$ is used to compute parameters of the approximate posterior which in our case was a normal distribution with diagonal covariance matrix, i.e. $q(\mathbf{z}|\mathbf{X}, \mathbf{x}, \boldsymbol{\phi}) = \mathcal{N}(\mathbf{z}|\mu(\mathbf{r}), \Sigma(\mathbf{r}))$.

A major difference between the generative matching networks and the originally proposed discriminative matching networks (Vinyals et al., 2016) is that since no label information is available to the model, the interpolation in equation (5) is performed not in the label space but rather in the prototype space which itself is defined by the model and is learnt during the training.

One can note that the described conditional model is not applicable in a situation where no conditioning objects are available. A possible solution to this problem involves implicit addition of a *pseudo-input* to the set of conditioning objects $\mathbf{X}$. A pseudo-input is not an actual observation, but rather just the corresponding outputs of functions $f$, $g$ and $\psi$ which are assumed to be another trainable parameters.

A stochastic computational graph describing the basic model with pseudo-input can be found on figure 1. Further by default we assume the presence of a single pseudo-input in the model and denote models without pseudo-input as **conditional**.

## 3.2 EXTENSIONS

Although the basic model is capable of instant adaptation to the conditioning dataset $\mathbf{X}$, it admits a number of extensions that can seriously improve it's performance.

The disadvantage of the basic matching procedure (5) is that conditioning observations $\mathbf{X}$ are embedded to the space $\Phi$ independently from each other. Similarly to discriminative matching networks we address this problem by computing *full contextual embeddings* (FCE) (Vinyals et al., 2015). In order to obtain a joint embedding of conditioning data we allow $K$ attentional passes over $\mathbf{X}$ of the form (5), guided by a recurrent controller $R$ which accumulates global knowledge about the conditioning data in its hidden state $\mathbf{h}$. The hidden state is thus passed to feature extractors $f$ and $g$ to obtain context-dependent embeddings.

We refer to this process as the full matching procedure which modifies equation (5) as:

$$\mathbf{r}_k = \sum_{t=1}^{T} a(\mathbf{z}, \mathbf{x}_t)\psi(\mathbf{x}_t), \quad a(\mathbf{z}, \mathbf{x}_t) = \frac{\exp(\text{sim}(f(\mathbf{z}, \mathbf{h}_k), g(\mathbf{x}_t, \mathbf{h}_k)))}{\sum_{t'=1}^{T} \exp(\text{sim}(f(\mathbf{z}, \mathbf{h}_k), g(\mathbf{x}_{t'}, \mathbf{h}_k)))}, \quad (6)$$
$$\mathbf{h}_{k+1} = R(\mathbf{h}_k, \mathbf{r}_k).$$

The output of the full matching procedure is thus the interpolated prototype vector from the last iteration $\mathbf{r}_K$ and the last hidden state of $\mathbf{h}_{K+1}$.

Besides context-dependent embedding of the conditioning data, full matching procedure allows to implement the data-dependent prior over latent variables $p(\mathbf{z}|\mathbf{X})$. In this case, no query point such as a latent variable $\mathbf{z}$ or an observation $\mathbf{x}$ is used to match with the conditioning data and only hidden state of the controller $\mathbf{h}$ is passed to functions $f$ and $g$. Output of the procedure is then used to compute parameters of the prior, i.e. means and standard deviations in our case.

As we discuss in the experiments section, we found these extensions so important that further we consider only the model with full matching described by equation (6) and data-dependent prior. Please refer to the appendix and the source code for architectural details of our implementation.

### 3.3 TRAINING

Training of our model consists of maximizing marginal likelihood of a dataset $\mathbf{X}$ which can be expressed as:

$$p(\mathbf{X}|\boldsymbol{\theta}) = \prod_{t=1}^{T} p(\mathbf{x}_t|\mathbf{X}_{<t}, \boldsymbol{\theta}), \quad \mathbf{X}_{<t} = \{\mathbf{x}_s\}_{s=1}^{t-1}. \quad (7)$$

Ideally we would like to use the whole available training data as $\mathbf{X}$ but due to computational limitations we instead use a training strategy rooted in curriculum learning (Bengio et al., 2009) and meta-learning (Thrun, 1998; Vilalta & Drissi, 2002; Hochreiter et al., 2001) which recently was successfully applied for one-shot discriminative learning (Santoro et al., 2016). In particular, we define a *task-generating* distribution $p_d(\mathbf{X})$ which in our case samples datasets $\mathbf{X}$ of size $T$ from training examples. Then we train our model to explain well all of the sampled datasets simultaneously:

$$\mathbb{E}_{p_d(\mathbf{X})}[p(\mathbf{X}|\boldsymbol{\theta})] \to \max_{\boldsymbol{\theta}}. \quad (8)$$

Obviously, the structure of task-generating distribution has a large impact on training and using an arbitrary distribution will unlikely lead to good results. Hence, we assume that at the training time we have an access to label information and can distinguish different concepts or classes. We thus constrain $p_d(\mathbf{X})$ to generate datasets consisting of examples that represent up to $C$ randomly selected classes so that even on short datasets the model has a clear incentive to re-use conditioning data. This may be considered as a form of weak supervision but we want to emphasize that one does not need the label information at test time unless the model is deliberately used for classification which is also possible.

Since the marginal likelihood (7) as well as the conditional marginal likelihoods are intractable we instead use variational lower bound (see section 2.1) as a proxy to $p(\mathbf{X}|\boldsymbol{\theta})$ in the objective (8):

$$\mathcal{L}(\mathbf{X}, \boldsymbol{\theta}, \boldsymbol{\phi}) = \sum_{t=1}^{T} \mathbb{E}_{q(\mathbf{z}_t|\mathbf{x}_t, \mathbf{X}_{<t}, \boldsymbol{\phi})} \left[\log p(\mathbf{x}_t, \mathbf{z}_t|\mathbf{X}_{<t}, \boldsymbol{\theta}) - \log q(\mathbf{z}_t|\mathbf{x}_t, \mathbf{X}_{<t}, \boldsymbol{\phi})\right].$$

## 4 EXPERIMENTS

For our experiments we use the Omniglot dataset (Lake et al., 2015) which consists of 1623 classes of handwritten characters from 50 different alphabets. The first 30 alphabets are devoted for training and the remaining 20 alphabets are left for testing. Importantly, only 20 examples of each class are available which makes this dataset specifically useful for small-shot learning problems. Unfortunately, the literature is inconsistent in usage of the dataset and multiple versions of Omniglot were used for evaluation which differ by train/test split, resolution, binarization and augmentation, see e.g. (Burda et al., 2015; Rezende et al., 2016; Santoro et al., 2016).

We use the canonical split provided by Lake et al. (2015). In order to speed-up training we down-scaled images to $28 \times 28$ resolution and since the result was fully binary we did not apply any further

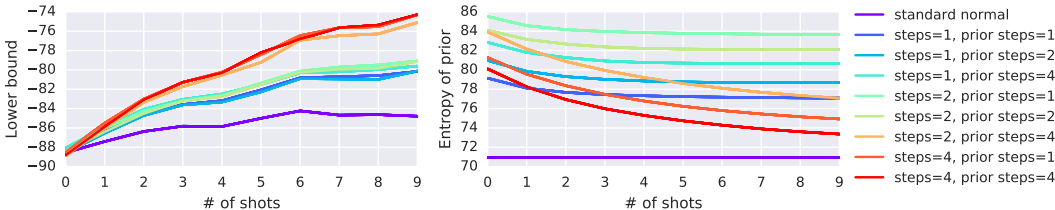

Figure 2: Lower bound estimates (left) and entropy of prior (right) for various numbers of attention steps and numbers of conditioning examples. Numbers are reported for the training part of Omniglot.

pre-processing. We also did not augment our data in contrast to (Santoro et al., 2016; Edwards & Storkey, 2016) to make future comparisons with our results easier.

Unless otherwise stated, we train models on datasets of length $T = 20$ and of up to $C = 2$ different classes as we did not observe any improvement from training with larger values of $C$.

### 4.1 NUMBER OF ATTENTION STEPS

Since the full context matching procedure (6) described in section 3.2 consists of multiple attention steps, it is interesting to see the effect of these numbers on model's performance. We trained several models with smaller architecture and $T = 10$ varying number of attention steps allowed for the likelihood and recognition shared controller and the prior controller respectively. The models were compared using exponential moving averages of lower bounds corresponding to different numbers of conditioning examples $\mathbf{X}_{<t}$ obtained during the training. Results of the comparison can be found on figure 2.

Interestingly, larger numbers of steps lead to better results, however lower bounds are almost not improving after the shared controller is allowed for 4 steps. This behaviour was not observed with discriminative matching networks perhaps confirming the difficulty of unsupervised learning. Another important result is that the standard Gaussian prior makes adaptation significantly harder for the model yet still possible which justifies the importance of adaptation not just for the likelihood model but also for the prior.

One may also see that all models preferred to set higher variances for a prior resulting to higher entropy comparing to standard normal prior. Clearly as more examples are available, generative matching networks become more certain about the data and output less dispersed Gaussians.

Based on this comparison we decided to proceed with models that have 4 steps for the shared controller and a single step for the prior controller which is a reasonable compromise between computational cost and performance.

### 4.2 FAST ADAPTATION AND SMALL-SHOT GENERATION

In this section we compare generative matching networks with a set of baselines by expected conditional likelihoods $\mathbb{E}_{p_d(\mathbf{X})} p(\mathbf{x}_t|\mathbf{X}_{<t})$. The conditional likelihoods were estimated using importance sampling with 1000 samples from the recognition model used as a proposal.

As we mention in section 3.1, it is possible to add a pseudo-input to the model to make it applicable for cases when no conditioning data is available. In this comparison by default we assume that a single pseudo-input was added to the model, otherwise we denote a model with no pseudo-input as **conditional**. When training and evaluating conditional models we ensure that the first $C$ objects in a dataset belong to different classes so that they in principle contain enough information to explain rest of the dataset.

We found it hard to properly compute conditional likelihoods for the neural statistician model (3) and hence had to exclude this model from the comparison, please see appendix for the details. Instead, we consider a simple generative matching network denoted as **avg** in which the matching procedure is replaced with prototype averaging which makes the adaptation mechanism similar to the one used in neural statistician. We also omitted sequential generative models (Rezende et al., 2016) from the

Table 1: Conditional negative log-likelihoods for the test part of Omniglot.

| Model | $C_{\text{test}}$ | Number of conditioning examples | | | | | | | |
|---|---|---|---|---|---|---|---|---|---|
| | | **0** | **1** | **2** | **3** | **4** | **5** | **10** | **19** |
| GMN, $C_{\text{train}} = 2$ | 1 | 89.7 | 83.3 | 78.9 | 75.7 | 72.9 | 70.1 | 59.9 | 45.8 |
| GMN, $C_{\text{train}} = 2$ | 2 | 89.4 | 86.4 | 84.9 | 82.4 | 81.0 | 78.8 | 71.4 | 61.2 |
| GMN, $C_{\text{train}} = 2$ | 3 | 89.6 | 88.1 | 86.0 | 85.0 | 84.1 | 82.0 | 76.3 | 69.4 |
| GMN, $C_{\text{train}} = 2$ | 4 | 89.3 | 88.3 | 87.3 | 86.7 | 85.4 | 84.0 | 80.2 | 73.7 |
| GMN, $C_{\text{train}} = 2$, conditional | 1 | | 93.5 | 82.2 | 78.6 | 76.8 | 75.0 | 69.7 | 64.3 |
| GMN, $C_{\text{train}} = 2$, conditional | 2 | | | 86.1 | 83.7 | 82.8 | 81.0 | 76.5 | 71.4 |
| GMN, $C_{\text{train}} = 2$, conditional | 3 | | | | 86.1 | 84.7 | 83.8 | 79.7 | 75.3 |
| GMN, $C_{\text{train}} = 2$, conditional | 4 | | | | | 86.8 | 85.7 | 82.5 | 78.0 |
| VAE | | 89.1 | | | | | | | |
| GMN, $C_{\text{train}} = 1$, avg | 1 | 92.4 | 84.5 | 82.3 | 81.4 | 81.1 | 80.4 | 79.8 | 79.7 |
| GMN, $C_{\text{train}} = 2$, avg | 2 | 88.2 | 86.6 | 86.4 | 85.7 | 85.3 | 84.5 | 83.7 | 83.4 |
| GMN, $C_{\text{train}} = 1$, avg, conditional | 1 | | 88.0 | 84.1 | 82.9 | 82.4 | 81.7 | 80.9 | 80.7 |
| GMN, $C_{\text{train}} = 2$, avg, conditional | 2 | | | 85.7 | 85.0 | 85.3 | 84.6 | 84.5 | 83.7 |

comparison as they were reported to overfit on the canonical train/test split of Omniglot. Another baseline we use is a standard variational autoencoder which has the same architecture for generative and recognition model as the full generative matching networks.

Table 1 contains results of the evaluation on the test alphabets from Omniglot. $C_{\text{train}}$ and $C_{\text{test}}$ denote the maximum number of classes in task-generating distributions $p_d(\cdot)$ used for training and evaluating respectively.

As one could expect, larger values of $C_{\text{test}}$ make adaptation harder since on average less examples of the same class are available to the model. Still generative matching networks are capable of working in low-data regime even when testing setting is harder than one used for training, i.e. $C_{\text{test}} > C_{\text{train}}$. Unsurprisingly, adaptation by averaging over prototype features performed reasonably well for simple datasets constructed of a single class, although significantly worse than the proposed matching procedure. On more difficult datasets with mixed examples of two different classes ($C_{\text{test}} = 2$) averaging was ineffective for expressing dependency on the conditioning data which justifies our argument on the necessity of nonparametric representations.

In order to visually assess the fast adaptation ability of generative matching networks we also provide conditionally generated samples in figure 3. Interestingly, the conditional version of our model which does not use a pseudo-input both at training and testing time generated samples slightly more similar to the conditioning data while sacrificing the predictive performance. Therefore, presence or absence of the pseudo-input should depend on target application of the model, i.e. density estimation or producing new examples.

## 5 CONCLUSION

In this paper we presented a new class of conditional deep generative models called generative matching networks. These models are capable of fast adaptation to conditioning dataset by adjusting both the latent space and the predictive density while making very few assumptions on the data. The nonparametric matching enabling these features can be seen as a generalization of the original matching procedure since it allows a model to define the label space itself extending the applicability of matching networks to unsupervised and perhaps semi-supervised settings. We believe that these ideas can evolve further and help to implement more data-efficient models in other domains such as reinforcement learning where data acquisition is especially hard.

ACKNOWLEDGMENTS

We would like to thank Michael Figurnov and Timothy Lillicrap for useful discussions. Dmitry P. Vetrov is supported by RFBR project No.15-31-20596 (mol-a-ved) and by Microsoft: MSU joint research center (RPD 1053945).

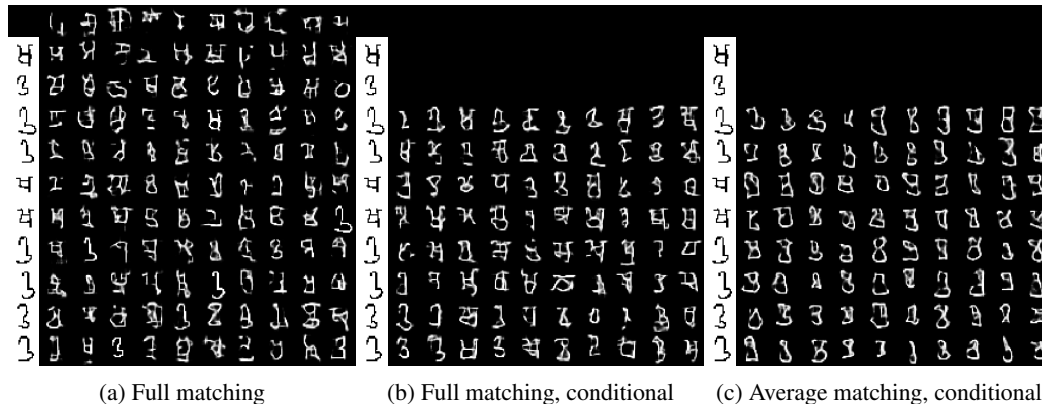

(a) Full matching (b) Full matching, conditional (c) Average matching, conditional

Figure 3: Conditionally generated samples. First column contains conditioning data in the order it is revealed to the model. Row number $t$ (counting from zero) consists of samples conditioned on first $t$ input examples.

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

## APPENDIX A. MODEL ARCHITECTURE

### CONDITIONAL GENERATOR

The conditional generator network producing parameters for $p(\mathbf{x}|\mathbf{z}, \mathbf{X}, \boldsymbol{\theta})$ has concatenation of $\mathbf{z}$ and the output of the matching operation $[\mathbf{r}, \mathbf{h}]$ as input which is transformed to $3 \times 3 \times 32$ tensor and then passed through 3 residual blocks of transposed convolutions. Each block has the following form:

$$h = \mathrm{conv}_1(x),$$
$$y = f(\mathrm{conv}_2(h) + h) + \mathrm{pool}(\mathrm{scale}(x)),$$

where $f$ is a non-linearity which in our architecture is always parametric rectified linear function (He et al., 2015).

The block is parametrized by size of filters used in convolutions $\mathrm{conv}_1$ and $\mathrm{conv}_2$, shared number of filters $F$ and stride $S$.

- scale is another convolution with $1 \times 1$ filters and the shared stride $S$.
- In all other convolutions number of filters is the same and equals $F$.
- $\mathrm{conv}_1$ and pool have also stride $S$.
- $\mathrm{conv}_2$ preserve size of the input by padding and has stride 1.

Blocks used in our paper have the following parameters $(W_1 \times H_1, W_2 \times H_2, F, S)$:

1. $(2 \times 2, 2 \times 2, 32, 2)$
2. $(3 \times 3, 3 \times 3, 16, 2)$
3. $(4 \times 4, 3 \times 3, 16, 2)$

Then log-probabilities for binary pixels were obtained by summing the result of these convolutions along the channel dimension.

Table 2: Conditional negative log-likelihoods for the test part of MNIST. Models were trained on the train part of Omniglot.

| Model | $C_{\text{test}}$ | \multicolumn{8}{c}{Number of conditioning examples} | | | | | | | |
|---|---|---|---|---|---|---|---|---|---|
| | | **0** | **1** | **2** | **3** | **4** | **5** | **10** | **19** |
| GMN, $C_{\text{train}} = 2$ | 1 | 126.7 | 121.1 | 118.4 | 117.6 | 117.1 | 117.1 | 117.1 | 118.5 |
| GMN, $C_{\text{train}} = 2$ | 2 | 126.2 | 123.1 | 121.3 | 120.1 | 119.4 | 118.9 | 118.3 | 119.6 |
| GMN, $C_{\text{train}} = 2$, conditional | 1 | | 135.1 | 120.9 | 117.5 | 115.7 | 114.4 | 111.7 | 109.8 |
| GMN, $C_{\text{train}} = 2$, conditional | 2 | | | 123.1 | 121.9 | 119.4 | 118.8 | 115.2 | 113.2 |
| GMN, $C_{\text{train}} = 1$, avg | 1 | 131.5 | 126.5 | 123.3 | 121.9 | 121.0 | 120.2 | 118.6 | 117.5 |
| GMN, $C_{\text{train}} = 2$, avg | 2 | 126.2 | 122.8 | 121.0 | 119.9 | 118.9 | 118.7 | 117.8 | 116.8 |
| GMN, $C_{\text{train}} = 1$, avg, conditional | 1 | | 132.1 | 126.9 | 125.0 | 124.8 | 123.9 | 121.7 | 120.9 |
| GMN, $C_{\text{train}} = 2$, avg, conditional | 2 | | | 118.4 | 117.9 | 117.4 | 117.1 | 116.6 | 115.8 |

FEATURE ENCODER $\psi$

Function $\psi$ has an architecture which is symmetric from the generator network. The only difference is that the scale scale operation is replaced by bilinear upscaling.

The residual blocks for feature encoder has following parameters:

1. $(4 \times 4, 3 \times 3, 16, 2)$

2. $(3 \times 3, 3 \times 3, 16, 2)$

3. $(2 \times 2, 2 \times 2, 32, 2)$

The result is a tensor of $3 \times 3 \times 32 = 288$ dimensions.

FUNCTIONS $f$ AND $g$

Each function $f$ or $g$ used in our model is simply an affine transformation of feature encoder's output (interpreted as a vector) to a 200-dimensional space followed by parametric rectified non-linearity.

## APPENDIX B. TRANSFER TO MNIST

In this experiment we test the ability of generative matching networks to adapt not just to new concepts, but also to a new *domain*. Since we trained our models on $28 \times 28$ resolution for Omniglot it should be possible to apply them on MNIST dataset as well. We used the test part of MNIST to which we applied a single random binarization.

Table 2 contains estimated predictive likelihood for different models. Qualitative results from the evaluation on Omniglot remain the same. Although transfer to a new domain caused significant drop in performance for all of the models, one may see that generative matching networks still demonstrate the ability to adapt to conditioning data. At the same time, average matching does not seem to efficiently re-use the conditioned data in such transfer task since relative improvements in expected conditional log-likelihood are rather small. Apparently, the model trained on a one-class datasets also learned highly dataset-dependent features as it actually performed even worse than the model with $C_{\text{train}} = 2$.

We also provide conditional samples on figure 4. Both visual quality of samples and test log-likelihoods are significantly worse comparing to Omniglot which can be caused by a visual difference of the MNIST digits from Omniglot characters. The images are bolder and less regular due to binarization. Edwards & Storkey (2016) suggest that the quality of transfer may be improved by augmentation of the training data, however for the sake of experimental simplicity and reproducibility we resisted from any augmentation.

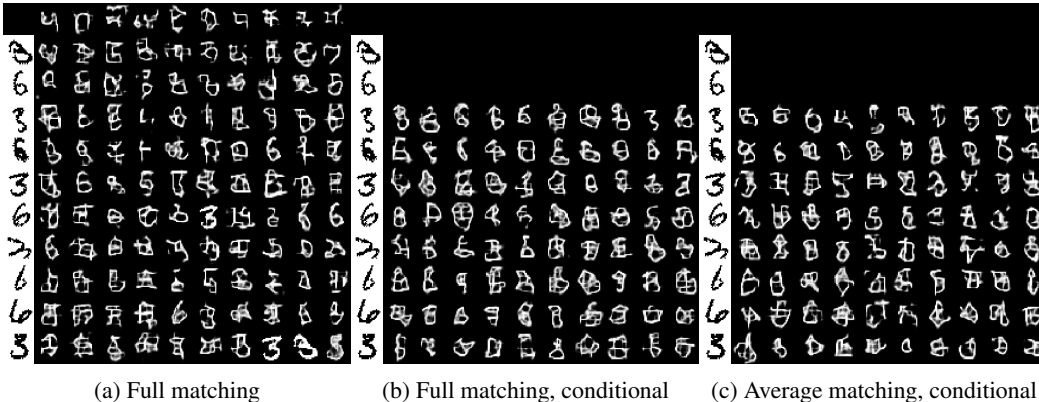

(a) Full matching                    (b) Full matching, conditional         (c) Average matching, conditional

Figure 4: Conditionally generated samples on MNIST. Models were trained on the train part of Omniglot. Format of the figure is similar to fig. 3.

Table 3: Small-shot classification accuracy (%) on the test part of Omniglot

| | | 5-way | | 20-way | |
|---|---|---|---|---|---|
| Model | Method | 1-shot | 5-shot | 1-shot | 5-shot |
| GMN, $C_{train} = 1$, conditional | likelihood | 82.7 | 97.4 | 64.3 | 90.8 |
| GMN, $C_{train} = 1$, avg, conditional | likelihood | 90.8 | 96.7 | 77.0 | 91.0 |
| GMN, $C_{train} = 1$, conditional | mean cosine | 62.7 | 80.8 | 45.1 | 67.2 |
| GMN, $C_{train} = 1$, avg, conditional | mean cosine | 72.0 | 86.0 | 50.1 | 72.6 |
| 1-NN, raw pixels | cosine | 34.8 | 50.5 | 15.6 | 28.2 |

## APPENDIX C. CLASSIFICATION

Generative matching networks are useful not only as adaptive density estimators. For example, one may use a pre-trained model for classification in several ways. Given a small number of labeled examples $\mathbf{X}_c = \{\mathbf{x}_{c,1}, \mathbf{x}_{c,2}, \ldots \mathbf{x}_{c,N}\}$ for each class $c \in \{1, 2, \ldots, C\}$, it possible to use the probability $p(\mathbf{x}|\mathbf{X}_c)$ as a relative score to assign class $c$ for a new object $\mathbf{x}$.

Alternatively, one may use the recognition model $q(\mathbf{z}|\mathbf{X}_1, \ldots, \mathbf{X}_C)$ to extract features describing the new object $\mathbf{x}$ and then use a classifier of choice, e.g. the nearest neighbour classifier. We implemented this method using cosine similarity on mean parameters of approximate Normal posteriors.

The results under different number of training examples available are provided in table 3. Surprisingly, the simpler model with average matching performed slightly better than the full matching model. Perhaps, generative matching networks are very smooth density models and even being conditioned on a number of same-class example still assign enough probability mass to discrepant observations. The same conclusion can be made by assessing the generated samples on figure 3 which may guide further research on the topic.

## APPENDIX D. EVALUATION OF THE NEURAL STATISTICIAN MODEL

The neural statistician model falls into the category of models with global latent variables which we describe in section 2.2. The conditional likelihood for these models has the form:

$$p(\mathbf{x}|\mathbf{X}, \boldsymbol{\theta}) = \int p(\boldsymbol{\alpha}|\mathbf{X}, \boldsymbol{\theta}) p(\mathbf{x}|\boldsymbol{\alpha}, \boldsymbol{\theta}) d\boldsymbol{\alpha}.$$

This quantity is hard to compute since it consists of an expectation with respect to the true posterior over global variable $\boldsymbol{\alpha}$. Since this distribution is intractable, simple importance sampling can not be used to estimate the likelihood. Thus, we tried the following strategies.

First, we used self-normalizing importance sampling to directly estimate $p(\mathbf{x}|\mathbf{X}, \boldsymbol{\theta})$ as

$$\hat{p}(\mathbf{x}|\mathbf{X}, \boldsymbol{\theta}) = \frac{\sum_{s=1}^{S} w_s p(\mathbf{x}, \mathbf{z}^{(s)}|\boldsymbol{\alpha}^{(s)}, \boldsymbol{\theta})}{\sum_{s=1}^{S} w_s}, \quad w_s = \frac{p(\boldsymbol{\alpha}^{(s)}, \mathbf{X}, \mathbf{Z}^{(s)}|\boldsymbol{\theta})}{q(\boldsymbol{\alpha}^{(s)}|\mathbf{X}, \boldsymbol{\phi})q(\mathbf{Z}^{(s)}, \mathbf{z}^{(s)}|\mathbf{X}, \mathbf{x}, \boldsymbol{\alpha}^{(s)}, \boldsymbol{\phi})},$$

but observed somewhat contradictory results such as non-monotonic dependency of the estimate on the size of conditioning dataset. The diagnostic of the effective sample size suggested that the recognition model is not well suited as proposal for the task.

Another strategy was to sequentially estimate $p(\mathbf{X}_{<t}, \boldsymbol{\theta})$ and then use the equation

$$p(\mathbf{x}_t|\mathbf{X}_{<t}, \boldsymbol{\theta}) = \frac{p(\mathbf{x}_t, \mathbf{X}_{<t}|\boldsymbol{\theta})}{p(\mathbf{X}_{<t}|\boldsymbol{\theta})},$$

which appeared to as unreliable as the previous strategy.

