# Peer review of "Fast Adaptation in Generative Models with Generative Matching Networks"

_ICLR 2017 — rejected_

[Official Review · AnonReviewer3 · rating 7 · confidence 4 · 16 Dec 2016]
**Interesting paper, good experiments**

The paper explores a VAE architecture and training procedure that allows to generate new samples of a concept based on several exemplars that are shown to the model. The proposed architecture processes the set of exemplars with a recurrent neural network and aggregation procedure similar to the one used in Matching Networks. The resulting "summary" is used to condition a generative model (a VAE) that produces new samples of the same kind as the exemplars shown. The proposed aggregation and conditioning procedure are better suited to sets of exemplars that come from several classes than simple averaging.
Perhaps surprisingly the model generalizes from generation conditioned on samples from 2 classes to generation conditioned on samples from 4 classes.
The experiments are conducted on the OMNIGLOT dataset and are quite convincing. An explicit comparison to previous works is lacking, but this is explained in the appendices, and a comparison to architectures similar to previous work is presented.

[Official Review · AnonReviewer2 · rating 4 · confidence 4 · 17 Dec 2016]
**reasonable approach, but the exposition is hard to follow, and it's not clear what is novel**

This paper presents a meta-learning algorithm which learns to learn generative models from a small set of examples. It’s similar in structure to the matching networks of Vinyals et al. (2016), and is trained in a meta-learning framework where the inputs correspond to datasets. Results are shown on Omniglot in terms of log-likelihoods and in terms of generated samples. 

The proposed idea seems reasonable, but I’m struggling to understand various aspects of the paper. The exposition is hard to follow, partly because existing methods are described using terminology fairly different from that of the original authors. Most importantly, I can’t tell which aspects are meant to be novel, since there are only a few sentences devoted to matching networks, even though this work builds closely upon them. (I brought this up in my Reviewer Question, and the paper has not been revised to make this clearer.)

I’m also confused about the meta-learning setup. One natural formulation for meta-learning of generative models would be that the inputs consist of small datasets X, and the task is to predict the distribution from which X was sampled. But this would imply a uniform weighting of data points, which is different from the proposed method. Based on 3.1, it seems like one additionally has some sort of query q, but it’s not clear what this represents. 

In terms of experimental validation, there aren’t any comparisons against prior work. This seems necessary, since several other methods have already been proposed which are similar in spirit.

[Official Review · AnonReviewer1 · rating 5 · confidence 3 · 17 Dec 2016]
**Interesting idea, but clarity issues make the paper difficult to follow**

This paper proposes an interesting idea for rapidly adapting generative models in the low data regime. The idea is to use similar techniques that are used in one-shot learning, specifically ideas from matching networks. To that end, the authors propose the generative matching networks model, which is effectively a variational auto-encoder that can be conditioned on an input dataset. Given a query point, the model matches the query point to points in the conditioning set using an attention model in an embedding space (this is similar to matching networks). The results on the Omniglot dataset show that this method is successfully able to rapidly adapt to new input distributions given few examples.

I think that the method is very interesting, however the major issue for me with this paper is a lack of clarity. I outline more details below, but overall I found the paper somewhat difficult to follow. There are a lot of details that I feel are scattered throughout, and I did not get a sense after reading this paper that I would be able to implement the method and replicate the results. My suggestion is to consolidate the major implementation details into a single section, and be explicit about the functional form of the different embedding functions and their variants.

I was a bit disappointed to see that weak supervision in the form of labels had to be used. How does the method perform in a completely unsupervised setting? This could be an interesting baseline.

There is a lack of definition of the different functions. Some basic insight into the functional forms of f, g, \phi, sim and R would be nice. Otherwise it is very unclear to me what’s going on.

Section 3.2: “only state of the recurrent controller was used for matching”, my reading of this section (after several passes) is that the pseudo-input is used in the place of a regular input. Is this correct? Otherwise, this sentence/section needs more clarification. I noticed upon further reading in section 4.2 that there are two versions of the model: one in which a pseudo input is used, and one in which a pseudo input is not used (the conditional version). What is the difference in functional form between these? That is, how do the formulas for the embeddings f and g change between these settings?

“since the result was fully contrastive we did not apply any further binarization” what does it mean for a result to be fully contrastive?

For clarity, the figures and table refer to the number of shots, but this is never defined. I assume this is T here. This should be made consistent.

Figure 2: why is the value of T only 9 in this case? What does it mean for it to be 0? It is stated earlier that T should go up to 20 (I assume #shot corresponds to T). It also looks like the results continue to improve with an increased number of steps, I would like to see the results for 5 and maybe 6 steps as well. Presumably there will come a point where you get diminishing returns.

Table 1: is the VAE a fair baseline? You mention that Ctest affects Pd() in the evaluation. The fact that the VAE does not have an associated Ctest implies that the two models are being evaluated with a different metric. Can the authors clarify this? It’s important that the comparison is apples-to-apples.

MNIST is much more common than Omniglot for evaluating generative models. Would it be possible to perform similar experiments on this dataset? That way it can be compared with many more models.

Further, why are the negative log-likelihood values monotonically decreasing in the number of shots? That is, is there ever a case where increasing the number of shots can hurt things? What happens at T=30? 40?

As a minor grammatical issue, the paper is missing determiners in several sentences. At one point, the model is referred to as “she” instead of “it”. “On figure 3” should be changed to “in figure 3” in the experiments section.

[Author Response · Sergey Bartunov · 13 Jan 2017]
**Update**

We would like to thank the reviewers for their efforts!

We received many useful comments that helped us to improve the text.
Using the provided feedback, we have edited the paper and uploaded a new version which, hopefully, addresses the issues raised in reviews.

We have also sent our response to all three reviews and would be glad to continue the discussion.

[Final Decision · Program Chairs · 06 Feb 2017]
**ICLR committee final decision**

This work extends variational autoencoders to adapt to a new dataset containing a small number of examples. While this work is promising, two of the reviewers had serious concerns about clarity. A new version of the paper has been submitted, however I still find it too hard to follow and would find it hard to accurately describe what was done having read the main body of the paper.